# Liquid water infiltration into a layered snowpack: evaluation of a 3D water transport model with laboratory experiments

Hiroyuki Hirashima[1], Francesco Avanzi[2,*], Satoru Yamaguchi[1]

[1]Snow and Ice Research Center, National Research Institute for Earth Science and Disaster Resilience, Suyoshi-machi,
Nagaoka-shi, Niigata-ken, 940-0821, Japan
[2]Department of Civil and Environmental Engineering, Politecnico di Milano, Milano, Italy
[*]Now at: Department of Civil and Environmental Engineering, University of California, Berkeley, Berkeley, USA

*Correspondence to*: Hiroyuki Hirashima (hirasima@bosai.go.jp)

**Abstract.** The heterogeneous movement of liquid water through the snowpack during precipitation and snowmelt leads to complex liquid water distributions that are important for avalanche and runoff forecasting. We reproduced the formation of capillary barriers and the development of preferential flow through snow using a three-dimensional water transport model, which was then validated using laboratory experiments of liquid water infiltration into layered, initially dry snow. Three-dimensional simulations assumed the same column shape and size, grain size, snow density, and water input rate as the laboratory experiments. Model evaluation focused on the timing of water movement, thickness of the upper layer affected by ponding, water content profiles and wet snow fraction. Simulation results showed that the model reconstructs relevant features of capillary barriers including ponding in the upper layer, preferential infiltration far from the interface, and the timing of liquid water arrival at the snow base. In contrast, the area of preferential flow paths was usually underestimated and consequently the averaged water content in areas characterized by preferential flow paths was also underestimated. Improving the representation of preferential infiltration into initially dry snow is necessary to reproduce the transition from a dry-snow-dominant condition to a wet-snow-dominant one, especially in long-period simulations.

## 1 Introduction

The heterogeneous movement of liquid water through the snowpack during precipitation and snowmelt leads to complex liquid water distributions that impact the snow structure through wet snow metamorphism. Furthermore, grain growth and subsequent changes in pore sizes and pore size distribution under wet conditions decrease snow strength (Wakahama, 1968; Raymond and Tsushima, 1979; Colbeck, 1983; Brun and Ray, 1987; Marsh, 1987; Brun et al. 1989; Lehning et al., 2002; Yamanoi and Endo, 2002; Ito et al., 2012) and can lead to wet snow avalanches (Kattelmann, 1984; Fierz and Föhn, 1994; Baggi and Schweizer, 2008; Mitterer et al. 2011; Mitterer and Schweizer, 2013; Takeuchi and Hirashima, 2013; Wever et al.,

2016a). Liquid water movement through the snowpack also controls the lag between rain events or snowmelt and water arrival at the snow base.

In the early theories of liquid water movement, capillary gradients in snow were usually neglected (Colbeck, 1972; Colbeck and Davidson, 1972; Colbeck, 1974a, 1974b, 1976; Dunne et al., 1976; Wankiewicz, 1978). For example, Marsh and Woo (1985) developed a model of flow channels but neglected the gradient term of capillary pressure. A two-dimensional (2D) model by Illangasekare et al. (1990) considered the gradient of capillary pressure, but focused on the effects of ice layers without considering the dependency of capillary pressure on grain size and density. A 2D model by Daanen and Nieber (2009) adopted a van-Genuchten model with dependence on grain size. For each of these models, the main cause of heterogeneous water movement was attributed to refreezing and ice layers. In porous media (e.g. soil), water can pond owing to capillary barrier, which consequently delays infiltration (e.g. Clifford and Stephen, 1998; Kämpf et al., 2003); however, water can also pond and consequently form preferential flow in layered snow, even when no ice layer forms (Waldner et al., 2004; Eiriksson et al., 2013; Katsushima et al., 2013; Avanzi et al., 2016).

Capillary barriers form owing to differences in the matrix potential between layers. Hirashima et al. (2010) replicated capillary barrier formation in the SNOWPACK model using parameters of matrix potential obtained from gravity drainage column experiments performed by Yamaguchi et al. (2010). Wever et al. (2014) incorporated the Richards Equation into the SNOWPACK model and obtained a good correlation with observed runoff. Wever et al. (2015) compared upGPR data with lysimeter data and showed that, even if the simulated waterfront did not arrive at the snow base, runoff was still initiated. This was interpreted to reflect the effect of preferential flow, which was not included in the model (Wever et al., 2015).

More recent studies have explicitly modelled preferential flow; for example, Katsushima et al. (2013) used laboratory experiments in vertically homogeneous snow to show that water entry suction, which in turn is related to grain size, affects the formation of preferential flow. On the basis of this work, Hirashima et al. (2014a) developed a three-dimensional (3D) water transport model for snowpack that is able to reproduce preferential flow as a function of water entry suction and validated it using the results of Katsushima et al. (2013). However, as snowpacks typically contain multiple layers of snow with different densities and grain sizes, simulations and laboratory experiments of water infiltration for different snow layers remain necessary. Furthermore, because simulation results for layered snow have not yet been validated using real data (Hirashima et al., 2013; Hirashima et al., 2014b), the accuracy of the model remains uncertain. Avanzi et al. (2016) performed infiltration experiments for multi-layered snowpacks with different combinations of grain size and infiltration rate and measured liquid water distribution, thickness of the capillary barrier, and arrival time. In this study, simulations of liquid water infiltration into layered snowpacks were performed by reproducing the laboratory experiments of Avanzi et al. (2016). The purpose of this study was: (1) to evaluate the accuracy of a 3D water transport model in reproducing infiltration patterns in layered snow; (2) to gain further insight into the 3D infiltration process into layered snow by comparing simulation results with data from laboratory experiments; and (3) to identify future avenues of development for 3D water transport schemes in snow.

Recently, a dual domain approach has been suggested to consider preferential flow effects in 1D (Wever et al., 2016b; Würzer et al. 2017). Similarly, Leroux and Pomeroy (2017) developed a 2D water transport model basing on the scheme of Hirashima et al. (2014a), but considering melt-freeze processes. Reproducing heterogeneous processes in a 1D or 2D model requires several assumptions. In natural snow, water flow shows lateral spreading, especially at capillary barriers, which creates complex 3D stratigraphic features at a grain/layer scale. Furthermore, when 3D preferential flow paths form in dry snow, wet snow area is proportional to the square of preferential flow size and inversely proportional to the square of the distance between paths (see Fig. S1). For a 2D simulation, wet snow area is, e.g., proportional to preferential flow size and inversely proportional to the distance between paths (see Fig. S1). Considering a 3D geometry can, therefore, help to define the necessary parameterizations of preferential flow effects needed to inform models with a reduced number of dimensions. Note that, while Leroux and Pomeroy's model also includes temperature and melt-freeze processes, this is not expected to play a role here as the validation experiments were performed under isothermal conditions.

## 2 Simulation method

### 2.1 Model

Details of the multi-dimensional water transport model are provided in Hirashima et al. (2014a). Models of liquid water movement in porous media use the Richards' equation and the Darcy-Buckingham law, which require knowledge of capillary pressure gradients and hydraulic conductivity. However, while the equation parameters depend on porosity, pore shape, pore connectivity, size distribution, and tortuosity, they are frequently estimated from a combination of snow density and grain size (Jordan et al., 2008).

In the three-dimensional model used here (Hirashima et al., 2014a), the relationship between capillary pressure, water content, grain size, and snow density (the so-called water retention curve) was determined based on gravity drainage column experiments performed by Yamaguchi et al. (2012). The relationship between saturated hydraulic conductivity, snow density, and grain size was estimated from the results of Calonne et al. (2012), who considered snow microstructure using the equivalent sphere radius estimated from specific surface area (SSA, instead of grain size). We considered grain size to be equal to equivalent sphere radius (Hirashima et al., 2014a) assuming grain shapes are round. If the grain shape is dendritic, an alternative method to estimate saturated hydraulic conductivity is necessary, including a simulation of SSA (Carmagnola et al., 2014). Unsaturated hydraulic conductivity was estimated using the van Genuchten-Mualem model (Mualem, 1976; van Genuchten, 1980). Water entry suction, which is necessary to reproduce preferential flow (Hirashima et al., 2014a), was measured and formulated as a function of grain size following the approach of Katsushima et al. (2013).

## 2.2 Comparative simulation

Hirashima et al. (2014a) performed infiltration simulations within columns with only one layer of snow. A number of multi-layer simulations were also tested (Hirashima et al, 2013, 2014b); however, they were performed in 2D and were not validated with observations. In this study, validation of the water transport model for layered snow was performed using observations of infiltration patterns performed using dye trace experiments (Avanzi et al., 2016). In these experiments, snow samples were prepared in a cold room at −20◦C using refrozen melt forms. Snow was packed in a cylindrical container composed of several acrylic rings; the height and diameter of the rings were 20 mm and 50 mm, respectively. Each sample was composed of two layers: the upper layer was 10 cm thick, the lower layer was either 8 or 10 cm thick (see Avanzi et al., 2016). Then, samples were moved to a second cold room at 0◦C and stored for at least 12 h to reach initial conditions of dry snow at 0◦C. All samples were characterized by a finer-over-coarser layering (i.e. the upper layer was created using a smaller grain size than the lower one), which aimed to reproduce capillary barriers. The three classes of snow grain size included fine (0.25–0.5 mm), medium (1.0–1.4 mm), and coarse (2.0–2.8 mm). While this definition is convenient for the scope of this study, it is not consistent with the International Classification proposed by Fierz et al. (2009). Three water input rates were considered: 10 mm/h, 30 mm/h, and 100 mm/h. In total, 9 experiments were performed (i.e., one for each grain size/input rate combination).

The 3D simulations had dimensions of 5, 5, and 20 cm in the x, y, and z directions, respectively. The voxels were 5 mm on the sides. Voxels of more than 2.5 cm from the central axis were treated as an impermeable wall, which ensured that the simulated shape was columnar. Snow densities, grain sizes, and rates of water supply were set to the same values as in the laboratory experiments. Grain size distributions were not measured: instead, for fine and medium snow we used the median grain sizes obtained by Katsushima et al. (2013) using the same sieves (0.41 and 1.5 mm, respectively). Grain size for coarse snow was determined assuming it to be two times the median medium grain size (2.9 mm). Note that grain size is expressed in two-digit accuracy, but the simulations were performed using a four-digit accuracy, as in Avanzi et al. (2016). As a measurement of horizontal structural heterogeneity, we estimated the standard deviation following the approach of Hirashima et al. (2014a), who estimated that the standard deviation of grain size is 20% of the median grain size (Katsushima et al., 2013). In this simulation, heterogeneity of snow density was not provided. As with the cases of laboratory experiments, grain size combinations in the simulation were fine-over-coarse snow (FC), fine-over-medium snow (FM), and medium over coarse snow (MC). Values of snow density and water supply rates are shown in Table 1.

The evaluation of simulations focused on the thickness of the ponding layer at the textural interface, on the liquid water distribution, on the wet snow fraction at different heights, and on different timings that are relevant for liquid water movement in snow. These include water arrival at the interface between layers, breakthrough of preferential flow in the lower layer, and arrival of liquid water at sample base. Data of liquid water content, wet snow fraction, and thickness of the ponding layer were measured by Avanzi et al. (2016), whereas timings were obtained from available video recordings of the experiments. A small difference (mean of 0.5 min, maximum of 3 min for FC1) was found between the arrival times from

video recordings and those in Avanzi et al. (2016); data from videos were used here for consistency with the other timings (see Table 2). The simulated timings of water arrival at the interface, entering the lower layer, and arrival at the snow base refer to the lowest elements in the upper layer, the top three elements in lower layer, and the lowest elements of the sample, respectively. The water content in the top three elements of the lower layer was used to determine the timing of breakthrough because preferential flow began immediately after the water content of one of these elements became larger than zero.

## 3 Simulation results

### 3.1 Water percolation through preferential flow paths and capillary barrier

Some images of the development of capillary barriers and preferential flow for FC1 are shown in Fig. 1. These figures show the front surfaces 20 s after the beginning of the experiment (a and e), at the arrival time of water at the interface between layers (b and f), at the time of breakthrough of preferential flow into the lower layer (c and g) and at the arrival time at the snow base (d and h). The simulation results showed faster than measured arrival of water at the boundary (Table 2), which implied an overestimation of vertical velocity in the model's preferential flow for this experiment (Fig. 1). In Fig. 1b and 1f, elapsed times were indeed 35 and 17 min in the laboratory experiment and simulation, respectively. One possible cause is the underestimation of the area of preferential flow path, which was also considered by Hirashima et al. (2014a). A smaller path area would increase conductivity because liquid water would be more concentrated and push water towards the boundary faster. After arrival at the boundary, we found that liquid water ponded above the boundary owing to a capillary barrier. In images from just before the formation of preferential flow in the lower layer (Fig. 1c and g), the elapsed time was 85 min in the laboratory experiment and 79 min in the simulation (relative difference of 7% of the measurement value: i.e. in good agreement).

The size of laboratory experiments was restricted by the time needed to prepare and perform each of them. Also, the diameter of samples was consistent with the thickness of similar experiments in soils (see e.g. Hill and Parlange, 1972), whereas Avanzi et al. (2017) showed that preferential flow may be intrinsically coupled with wet-snow metamorphism at grain scale. This suggests that small-scale experiments are appropriate to understanding the physics of this process in snow. Nonetheless, the relatively small scale of these experiments may introduce some domain size effect. In natural snow, water flow shows lateral spreading, especially at capillary barriers, whereas experiments with small size may partially perturb the natural flow on snow and therefore change vertical flow owing to artificial edges. This may increase the ratio of preferential flow path area, decrease the arrival time at the base, and decrease natural ponding amount at the capillary barrier. In terms of comparison between simulations and experiments, this effect was offset by using the same domain conditions.

The times of liquid water arrival at the base following the formation of preferential flow through the lower layer were 4 and 1 min in the laboratory experiment and simulation, respectively. On this basis, we calculated the propagation rate of the preferential flow path to be 0.4 and 1.6 mm/s for the laboratory experiment and simulation, respectively.

In the other experiments, the temporal dynamics of preferential flow formation and water ponding at the interface were generally well reproduced (Table 2; Fig. 2). The root mean square error (RMSE), the slope of a regression line with intercept equal to 0, and the correlation coefficient ($r^2$) between the simulated and measured timings were 7.8 min., 0.97 and 0.93, respectively. As timings were measured using frontal movies, we were sometimes unable to evaluate the timing of preferential flow formation within a sample. For example, in the case of MC1, preferential flow formed on the side of the sample that was not visible from the frontal position of the camera. Thus, the frontal movie did not show the preferential flow path in lower layer, whereas horizontal spreading of water at the sample base allowed us to detect the arrival time with a reasonable precision. It follows that for this sample we cannot determine the timing of preferential flow initiation. Therefore, estimated timings from laboratory experiments may contain a delay. Overall, simulated and measured timings were consistentl, which confirms that if snow parameters (e.g. snow density and grain size) are known, the arrival time of liquid water can be predicted using this model.

### 3.2 Thickness of the water ponding layer

Avanzi et al. (2016) measured the thickness of the upper layer affected by ponding at the end of each experiment. In their results for FC and FM experiments, the liquid water content on the layer boundary was about 33% to 36% (2-cm vertical resolution). The volume of ponded water was smaller for MC experiments. Laboratory experiments also showed that the thickness of the water ponding layer is not strongly connected to the water input rate. In our simulations, the influence of water input rate on the thickness of the water ponding layer was also small; however, the influence of grain size was significant (Table 3). The thickness of ponded water at the interface was well reproduced for the FC experiments, but was overestimated for FM experiments and underestimated for MC experiments. For MC experiments, up to 1 cm of ponding was shown in laboratory experiments, while simulated results showed a thickness of less than 0.5 cm.

### 3.3 Horizontal cross section

During laboratory experiments, Avanzi et al. (2016) measured wet snow fractions at the boundary between consecutive rings using photos of the top surface of the ring below the boundary. Samples were likely slightly compressed during experiments owing to increased densification caused by wetting (Marshall et al. 1999), even though this was not noticeable. Because the model does not include settling, we chose to compare data with simulations of the inferior surface of the ring above the boundary, which returned more consistent results. At the interface between layers, most of the area of each section was wet, except for the medium over coarse samples (Fig. 3). For the other sections, only a fraction was found to be wet. This pattern was well simulated (Fig. 4); although simulated wet snow areas were smaller than those measured especially in areas characterized by preferential flow. Similar underestimation by the model was also observed by Hirashima et al. (2014a).

**3.4 Water content distribution**

Our simulations were performed with 5 mm voxels. Simulated water contents from all voxels at a given height were averaged to obtain the water content profile. In laboratory experiments, water content profiles were obtained with a resolution of 2 cm (Fig. 5). The results showed that for the FC and FM experiments, the liquid water content was overestimated near the interfaces between snow layers in the upper fine layer but underestimated in other areas. The impact of water supply rate on the water content in capillary barriers was small in both simulations and experiments. Overall, simulations and observations showed good agreement in that liquid water content increased with depth in the finer layer, peaked at the interface between layers, and decreased in the lower layer.

**4 Discussion**

**4.1 Comparison with SNOWPACK**

The numerical snowpack model SNOWPACK can also be used to reproduce dynamics observed during laboratory experiments. While Avanzi et al. (2016) compared their results with SNOWPACK-3.3.0 simulations at the end of each experiment (i.e. at the observed/modelled arrival time of water at the snow base), a direct comparison between models can be made for any point of time. Here, we compared temporal changes in the simulated water content profiles for SNOWPACK and the 3D model in order to assess the role played by a simulation of preferential flow in controlling liquid water distribution in snow (Fig. 6; Fig. S1); therefore, we used both the matrix-flow multi-layer implementation of the Richards Equation, RE-model (Wever et al., 2015), and a dual domain approach considering preferential flow, henceforth DDA-model (Wever et al., 2016b; Würzer et al, 2017). Resolution of SNOWPACK was set to 5 mm to match with the resolution of the 3D model.

In the SNOWPACK RE-model simulations, liquid water content in the upper layer gradually increased with time at all positions (Fig. 6a; Fig. S2), and the water content near the boundary was relatively large. The difference in water content between the layer interface and the upper part was underestimated when compared with experimental results, which confirms a marked spatial heterogeneity in liquid water distribution. On the other hand, 3D simulation showed that liquid water quickly ponds at the boundary, which is consistent with the experimental observations (Fig. 6e and f; Fig. S4). Such an effect is obtained owing to preferential flow, which allowed water to move in small fingers and to reach deeper locations, even when most of the upper snow remained dry. The water ponding layer thickened until the formation of a preferential flow path in the lower layer. After preferential flow arrived at the snow base, expansion of the water ponding layer stopped. The difference in water content between layer interface and the upper part was overestimated in comparison with the experimental observations. In case of the SNOWPACK DDA-model (Fig. 6c and d), liquid water arrived quickly at the

boundary and started to pond. Then, infiltration in matrix flow started. During ponding, infiltration into the lower layer was started in preferential flow areaa with a very small water content (about 0.01% initially, and then gradually increasing). Although liquid water arrival at the snow base was faster than that in the RE-model, water ponding continued even after the liquid water arrival because the infiltration rate was too small in the lower layer. After a large amount of water ponded above the layer boundary, liquid water infiltration in matrix flow in the lower layer was started and the volume of water ponded in the upper layer started to decrease (e.g. in FM3, water content in upper layer at 2t was decreased from 5/3t; Fig. S3d). The RMSE for liquid water content profiles at (measured) arrival time are 0.107, 0.107 and 0.094 for SNOWPACK RE-model, DDA-model and 3D model, respectively. While both SNOWPACK schemes yield the same RMSE, the 3D model returns a slightly smaller value.

For arrival times, the 3D and SNOWPACK DDA-model obtained greater accuracy than the SNOWPACK RE-simulations (Fig. 7), which again suggests the importance of considering preferential flow. Causes of delay in 1D models include both slow infiltration of matrix flow and overestimation of water ponding at the capillary barrier (Fig. 6). In terms of arrival time, delay was resolved to some degree by considering preferential flow in the SNOWPACK model. However, SNOWPACK-DDA-model is still prone to overestimating the total amount of water that pond at the interface between layers. These comparisons between the SNOWPACK, 3D model and laboratory experiments demonstrate the need to improve existing theories of water infiltration in snow.

The theory of water transport in the SNOWPACK RE-model is based on gravity drainage column experiments that neglect water entry suction (i.e. experiments performed using wet snow; Yamaguchi et al., 2012). In contrast, the 3D model and SNOWPACK DDA-model includes an attempt to simulate the infiltration process into initially dry snow using water entry suction (where we define dry snow as that with a lower liquid water content than the irreducible water content), which is key to reproducing fingers (Hirashima et al., 2014a). Under these conditions, the van Genuchten model could only be used with additional assumptions (Hirashima et al., 2014a). Accordingly, we assumed that dry snow had a threshold suction equal to water entry suction. Future work will focus on improving this approach; for example, water entry suction may be related to the suction–wetness profile of a wetting water retention curve (Avanzi et al., 2016), which has not yet been parameterized. Furthermore, unsaturated conductivity tends towards zero in dry conditions, but extensive observations of unsaturated conductivity in snow are missing.

## 4.2 Wet snow ratio and preferential flow path area.

The main purpose of the development of this model is to better understand 3D patterns of water infiltration in snow and, thus, resolve the delay of the arrival time as a limitation of matrix-flow models. The simulation results showed that the model can reproduce preferential flow and capillary barriers and, consequently, provide reliable estimations of the arrival time of water at the sample base. On the other hand, the model underestimated the simulated preferential flow area. In terms of effect on arrival time, this underestimation is not a serious problem because the travel time through the preferential flow area was

short (Table 2); however, it may represent a problem for long-term simulations, especially when estimating the transition from a predominantly dry snow to a predominantly wet snow.

According to the simple model of Baker and Hillel (1990), the wetted fraction of the sublayer in a finer-over-coarser transition depends on water input rate and unsaturated conductivity during steady vertical infiltration. Horizontal expansion of preferential flow also depends on infiltration along the horizontal direction. As the direction of water flow depends on gravity (vertical) and capillarity, movement in the horizontal direction may be impeded if simulated capillary gradients are small. For example, the fact that fine snow in experiments had larger preferential flow paths than coarse snow was probably due to a greater heterogeneity in capillarity in fine snow (Avanzi et al., 2016).

We performed sensitivity tests to estimate the relevance of vertical and horizontal movement for different types of snow, in which we calculated which voxel (left, right, front, back, up, down) was easiest to infiltrate from a generic voxel as a function of gravity or water entry suction. We found that the ratios of the water moving to the lower voxel were 24.3%, 38.8% and 60.7% for fine, medium and coarse snow, respectively. When this ratio is large (e.g. coarse snow), water moves downward, and consequently the preferential flow path areas become small. Where there is no gravitational force, the ratio would be 16.7% while for fine snow the ratio of water moving to the lower voxel was 24.3%. Nevertheless, the simulated mean wet snow area was small even for fine snow (e.g. 4.8% in FC1 and 22% in FC3, excluding the ponding area). As the simulated wet snow area is smaller than the measured one, this model may still underestimate the effective cross-sectional area of infiltration. This will be the subject of future research.

In this model, water entry suction was used as a threshold for liquid water infiltration into dry snow. However, in the measured water absorption curve of Adachi et al. (2012), the relationship between suction and liquid water content was non-linear and hysteretic (see Section 4.1). This simplified condition for infiltration into dry snow may lead to an underestimation of the expansion of preferential flow. Neglecting quick metamorphism in preferential flow paths (Avanzi et al. 2017) may represent another cause of underestimation of preferential flow path size as grain growth promotes lateral spreading of water and expansion of paths. Although this model includes grain growth following Brun et al. (1989) and Tusima (1977), modelling some specific conditions such as wet snow metamorphism at the boundaries between preferential flow paths and drier snow is still an open issue. Also, existing observations of wet-snow metamorphism have been mainly performed in static conditions, which means that the coupling between grain growth and flowing water is still poorly understood. This represents a further unknown for models of liquid water in snow.

The number of preferential flow paths can also promote the expansion of the wet snow area (Schneebeli, 1995). In our model, liquid water preferred to infiltrate snow along the same path; therefore, preferential flow paths did not increase unless the amount of liquid water supply also increased. Also, compaction by wet snow metamorphism could change the balance of force distribution and create new pathways for liquid water. This underestimation may also be related to uncertainties in the computation of unsaturated water conductivity in initially dry snow and/or in the rule used to calculate the conductivity between voxels. New techniques to measure the development of preferential flow paths can help to model these processes and further experiments in this direction are, therefore, highly needed.

### 4.3 Impact of grain size on water infiltration process

Grain size is one of the key parameters for water infiltration process. Nevertheless, grain sizes cannot be measured at high resolution in nature (typically measured to 0.1mm); therefore, the sensitivity of model results with fluctuation in grain size is informative. In this discussion, sensitivity experiments for grain sizes fluctuating by 0.1mm for the upper layer and lower layers were performed. Simulation results of in terms of thickness of water ponding layer, water content profile, and arrival time are shown in Fig. 8.

The thickness of water ponding over the interface between layers is usually increased in cases of smaller upper grain size and greater lower grain size (Fig. 8a). Grain size thus significantly influences the water ponding layer. Differences in water ponding thickness between an real and decreased grain size on the upper layer ranged from 7% to 48% in case of fine over coarse and fine over medium. On the other hand, differences for fluctuations of the lower layer were less than 8% in eight out of nine cases (Fig. 8a), sample MC1 being the only exception (22%). This difference suggests that the upper layer plays a pivotal role in capillary barriers. An increase in the water ponding layer in the case of decreased upper fine grains can also be clearly identified in the water content profile (Fig. 8b). Decreasing the size of upper fine grains not only increases the thickness of the water ponding layer, but also increases water content outside of the water ponding layer. For example, decreasing upper layer grain size in FC1 (Fig 8b) yields a total water contents from 15.5–19.5 cm height that is about 2.6 times those of the original, non-fluctuating case. This reflects the increasing area of wet snow in the upper layer. The area of the preferential flow path is sensitive to both grain sizes, especially for fine snow (Katsushima et al, 2013; Hirashima et al., 2014). Fluctuations in water content profile driven by grain size were also seen in case of MC1, but were not significant (see Fig. 8c). Figures of water content profiles for other cases are shown in the supplement (Fig. S5).

Variations in grain size also affect arrival time at the snow base (Fig. 8d). Differences in arrival time between increasing grain size and decreasing grain size for the upper layer was more than 50% in the cases of FC1 and FM1. These results suggest that the accuracy of measured grain size is important to estimate water infiltration, especially for fine grains. In the parameterizations for water entry suction used in our 3D model and dual domain approach, grain size was determined circumstantially providing a grain size distribution measured from individual grains. Since measuring individual grain size requires great care, methods requiring less care and the ability to obtain grain size parameters with high degrees of accuracy are necessary for more accurate models. Development of measuring methods for specific surface area may improve estimation accuracy and contribute the study of water infiltration processes.

### 4.4 Outlook

The 3D model developed here represents an important stage in the development of an exhaustive theory of liquid water movement in snow. However, the low accuracy of preferential flow path area in our model means that it cannot be used to

improve the parameterization of preferential flow area, as in Wever et al. (2016b). In the future, a thorough parameterization of hysteresis in snow and the better reconstruction of the expansion of preferential flow path area will improve the accuracy of 3D models and allow for an advanced estimation of preferential flow area in 1D models. Wever et al. (2016b) also suggested that 3D models should analyse heat exchange around the preferential flow path; therefore, future developments of our model will consider heating and melt-freeze processes (e.g. the model of Leroux and Pomeroy, 2017). For this, laboratory experiments of ice layer formation will be needed for validation.

Another possible improvement to the model would be the parameterization of quick grain growth at saturation, which would be necessary for simulating the structural evolution of areas affected by ponding. Grain growth causes an abrupt decrease in suction and consequently reduces the water content at the ponding layer. The water content of the upper layer at 2t was smaller than that at 5t/3 (Fig. 6a) owing to a decline of suction due to grain growth. In the first version of the SNOWPACK model, the Brun et al. (1989) equation was used for estimating grain growth; however, this formula is based on data with small water content and application to saturated conditions may overestimate grain growth. To avoid this, Hirashima et al. (2010, 2014a), used the equation of Tusima (1978) to constrain the upper limit of grain growth rate. Although this formula is based on data measured under saturated conditions, grain growth remains limited for a short time scale, such as occurred during these experiments. The grain growth equation of Tusima (1978) was formulated using data for 200 hours, but it did not focus on the first 1 hour; therefore, grain growth over short time periods and under saturated conditions remain unclear. Raymond and Tusima (1979), Wakahama (1968), and Colbeck (1973) all focused on wet snow metamorphism under saturated conditions, but they did not focus on the first hour of the experiments. Extending the existing parameterizations of wet snow metamorphism for small timescales will improve simulation accuracy with regards to the development and disappearance of water ponding by capillary barriers. Also, a correct simulation of grain growth will lead to correctly estimating the lateral spreading of water, which will improve the accuracy of the prediction of preferential flow path size.

Our results show that this model is capable of reproducing detailed water infiltration at sample scale (i.e., considering micro-scale heterogeneity). On the other hand, the intrinsic scale of this process and computational efforts mean that it is still not suitable for basin-scale simulations. This limitation could be overcome by synergies with existing physics-based hydrologic models for snow-dominated catchments; for example, Alpine 3D (Lehning et al., 2006). Currently, SNOWPACK is used as a part of Alpine3D for simulation of accumulation/ablation patterns of snowpack. In this study, comparisons between laboratory experiments, a 3D model, and SNOWPACK were performed and contributed to highlighting model limitations and possible avenues of future developments (e.g. an underestimation of flow path cross-sections). While a 3D model cannot reproduce the entire range of natural variability of liquid water flow in snow, it can help to replicate and understand this process in conditions that are difficult for experiments (e.g., larger sample sizes and/or a more complex stratigraphy). This may contribute to defining new parameterizations for dual domain approaches that could be then fully included in catchment-scale models. Also, we will try to apply this model at the basin scale by increasing the element size. While this will hamper the representation of single preferential fingers, we expect the model to be able to correctly reproduce other

relevant features of water flow at slope scale such as lateral flow. This could help to understand liquid water flow around concave/convex portions of the landscape.

## 5 Conclusions

Validation of simulations for capillary barrier formation and subsequent preferential flow development was performed using a 3D water transport model. Overall, the infiltration process into dry snow was well reproduced, and in particular the timing of liquid water arrival at the snow base was accurate. A detailed comparison of wet conditions in the snow column was performed to check accuracy and identify shortcomings in the model. The model accurately reproduced: (a) the onset of preferential flow in initially dry snow; (b) the ponding of liquid water above the boundary of snow layers by a capillary barrier, for which the ponded water volume was larger at the boundary of fine over coarse and fine over medium snow layers than it was at the boundary of medium over coarse snow layers.

Model discrepancies included: (a) an underestimation of liquid water content and wet snow area in preferential flow path areas, and (b) overestimation of water ponding volume at the layer boundary in experiments FC and FM, but underestimation in experiment MC. Future improvements to the model will include improving the water entry process for dry snow, measurements of water content profile for capillary rise, and direct measurements of preferential flow path formation.

The advantage of this model over 1D models is the consideration of 3D heterogeneous infiltration into dry snow. An explicit simulation of preferential flow also returns a reliable estimation of liquid water arrival at the snow base. However, improvements are needed to ensure that the model works over both long and short time periods. An accurate reproduction of the transition from a dry-snow dominant to a wet-snow dominant condition is an important step in upgrading this model to a full 3D numerical snowpack model.

## Acknowledgement

This study was part of the 'Research on Advanced Snow Information and its Application to Disaster Mitigation' project, and was supported by JSPS, KAKENHI (Grant Numbers JP15H01733). Fruitful discussions about this work with Y. Ishii and T. Katsushima are acknowledged. We would like to acknowledge N. Wever, who helped with the SNOWPACK simulations with preferential flow. We thank the members of the Snow and Ice Research Center for their advice and discussion. We are grateful to M. Miura, who assisted with our research.

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

**Table 1: Experimental conditions after Avanzi et al. (2016)**

| Sample ID | W (mm h$^{-1}$) | $\rho_{DU}$ (kg m$^{-3}$) | $\rho_{DL}$ (kg m$^{-3}$) |
|---|---|---|---|
| FC1 | 11.9 | 417 | 465 |
| FC2 | 28 | 449 | 483 |
| FC3 | 113 | 433 | 470 |
| FM1 | 11.9 | 444 | 484 |
| FM2 | 27.7 | 442 | 487 |
| FM3 | 110 | 455 | 510 |
| MC1 | 11 | 472 | 487 |
| MC2 | 27.3 | 498 | 480 |
| MC3 | 111 | 494 | 478 |

**Table 2: Timings of infiltration in laboratory experiments and simulations [a]**

| experiment ID | experiment | | | simulation | | |
|---|---|---|---|---|---|---|
| | arrival at boundary | preferential flow formation | arrival at snow base | arrival at boundary | preferential flow formation | arrival at snow base |
| FC1 | 34.8 | 85.0 | 89.0 | 16.7 | 79.0 | 79.7 |
| FC2 | 15.2 | 48.5 | 49.8 | 8.7 | 38.7 | 39.0 |
| FC3 | 7.1 | 12.3 | 14.0 | 4.0 | 11.7 | 12.0 |
| FM1 | 20.0 | 79.0 | 89.5 | 17.0 | 72.3 | 109.0 |
| FM2 | 11.3 | 33.3 | 39.8 | 10.7 | 37.3 | 58.3 |
| FM3 | 6.7 | 11.2 | 13.0 | 4.3 | 11.3 | 15.7 |
| MC1 | 5.3 | - | 9.5 | 9.0 | 11.0 | 11.7 |
| MC2 | 3.0 | 5.0 | 8.0 | 4.7 | 5.3 | 5.3 |
| MC3 | 0.8 | 2.5 | 4.5 | 1.7 | 1.7 | 1.7 |

a.  All the timings were estimated from the video recording of the experiment. The '-' represents points where timings could not
5      be estimated from the video recording.

**Table 3: Thickness of upper layer affected by ponding at the layer boundary[a]**

| Sample ID | experiment (cm) | simulation (cm) |
|-----------|-----------------|-----------------|
| FC1 | 2-3 | 2.8 (±0.5) |
| FC2 | 3-4 | 3.3 (±0.5) |
| FC3 | 2-3 | 4.1 (±0.4) |
| FM1 | 2-3 | 4.0 (±0.3) |
| FM2 | 2-3 | 5.1 (±0.3) |
| FM3 | 1-2 | 4.8 (±0.4) |
| MC1 | 0-1 | 0.1 |
| MC2 | 1-1 | 0.3 |
| MC3 | 0.5-1 | 0.3 |

**a. For each pixel at the interface between layers, simulated thickness was first determined by computing the number of voxels above with liquid water content (LWC) of >10%. These data were then used to calculate a mean value and its standard deviation.**

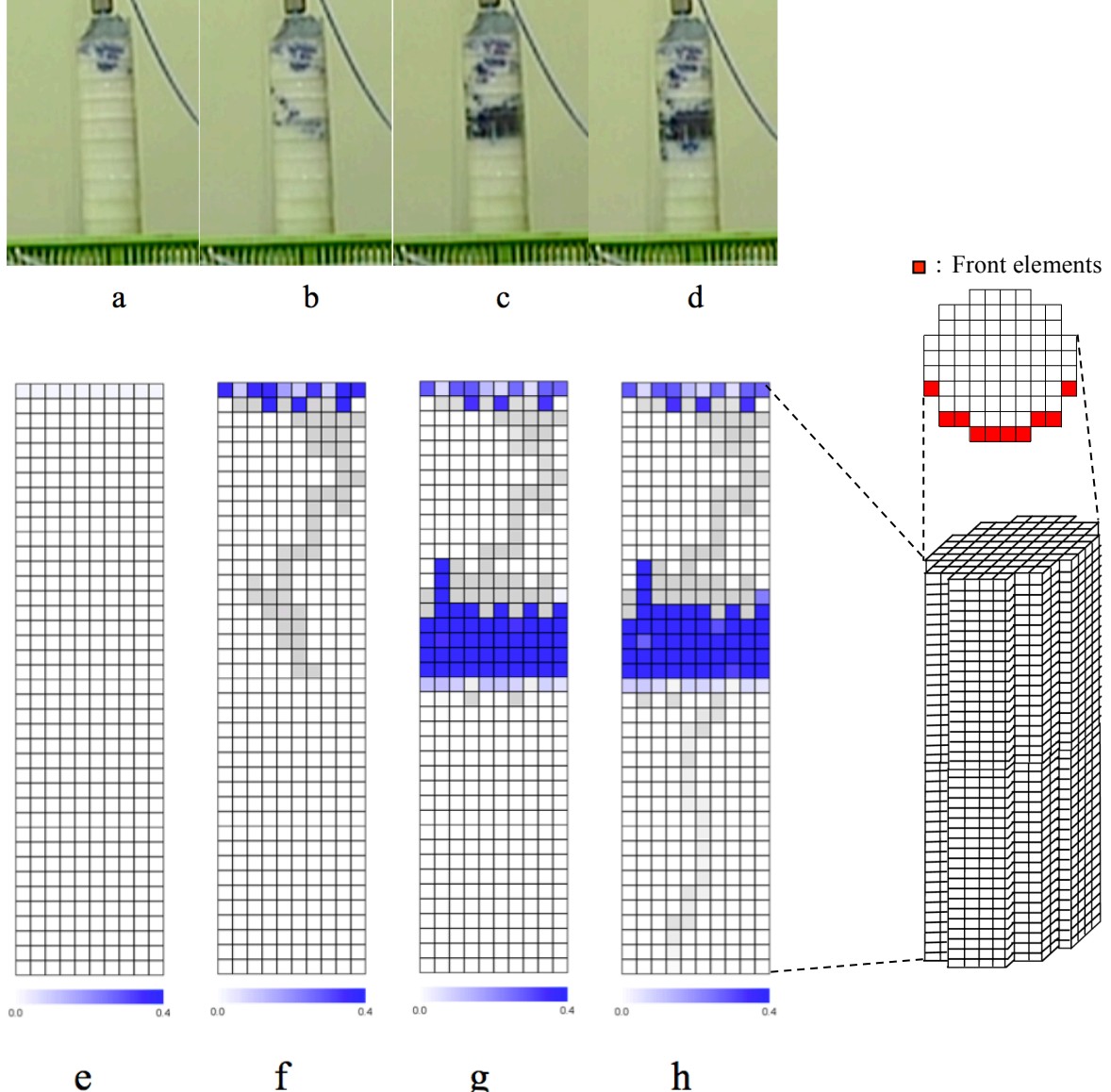

**Figure 1:** Development of capillary barrier and preferential flow path for FC1 during experiments (a–d) and simulations (e–h). A blue dye tracer was used in the experiment. In the simulation images, blue denotes the liquid water content at the front elements (see: right figure), while grey denotes that the front elements are dry, but some liquid water is present within the sample at that position. The grey-scale represents the maximum liquid water content for each location. Captured times were at: (a) 20 s, (b) 35 min, (c) 1 hour 25 min, (d) 1 hour 29 min, (e) 20 s, (f) 17 min, (g) 1 hour 19 min, and (h) 1 hour 20 min.

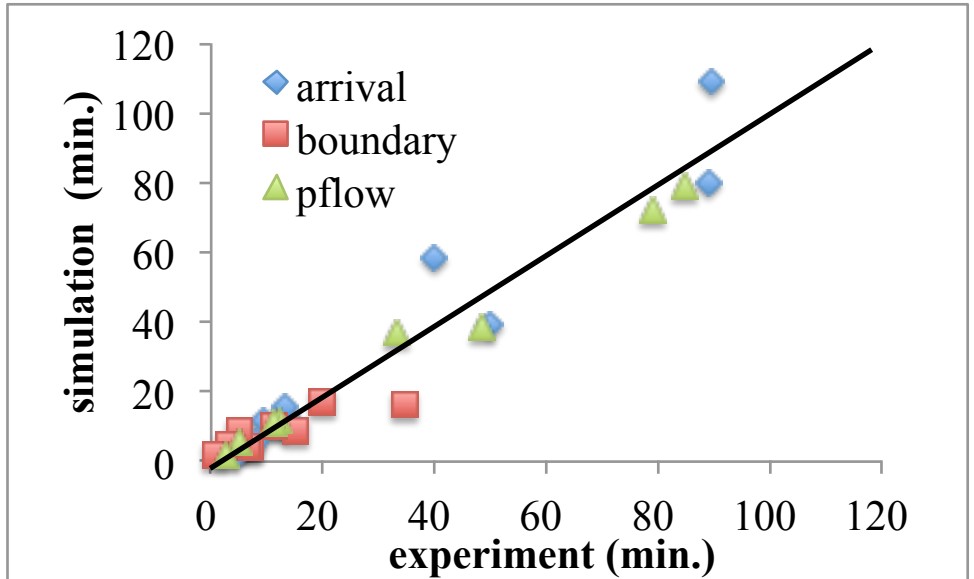

**Figure 2: Timings of water arrival at the interface between layers (red squares), formation of preferential flow path (green triangles), and arrival at snow base (blue diamonds) for experiments and simulations.**

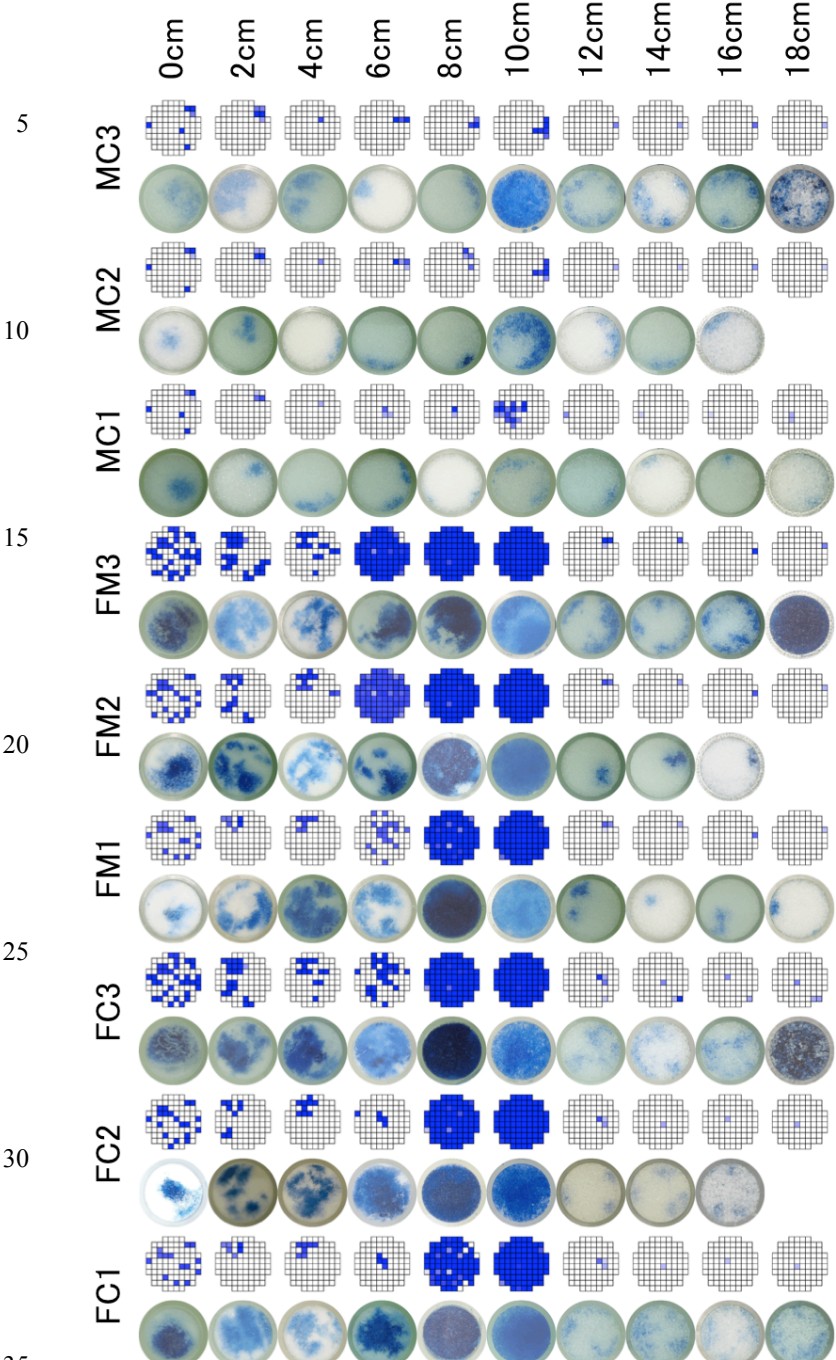

**Figure 3: Liquid water distribution (blue shading) at the end of each experiment and simulation. The coordinate on the right denotes the depth of the section from the top surface.**

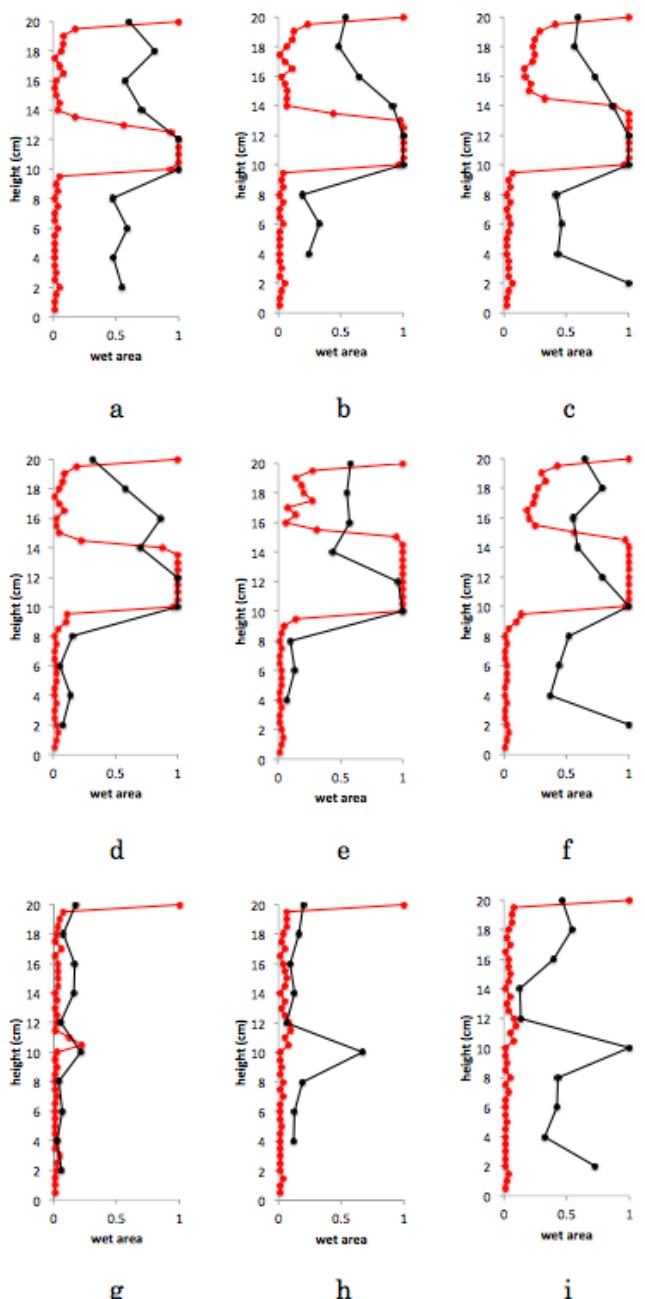

**Figure 4: Profiles of wet area for experiments (red line) and simulations (black line): (a) FC1, (b) FC2, (c) FC3, (d) FM1, (e) FM2, (f) FM3, (g) MC1, (h) MC2, and (i) MC3.**

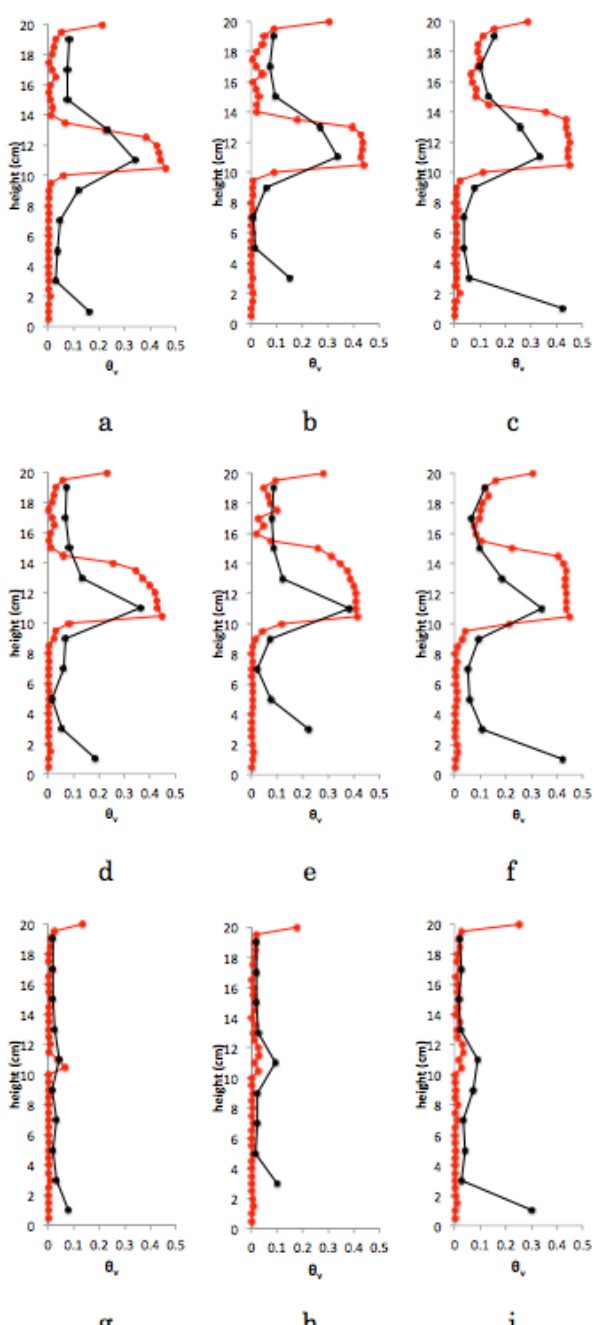

**Figure 5: Profiles of volumetric water content for experiments (red line) and simulations (black line): (a) FC1, (b) FC2, (c) FC3, (d) FM1, (e) FM2, (f) FM3, (g) MC1, (h) MC2, (i) MC3.**

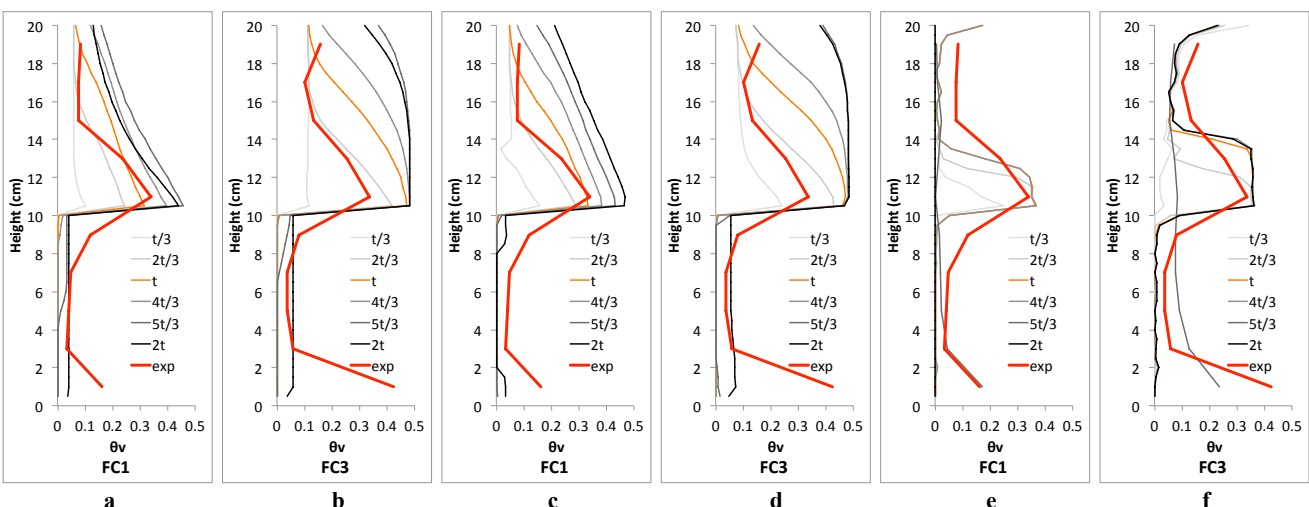

5    **Figure 6: Temporal evolution of simulated water content profiles: (a) FC1 in SNOWPACK RE-model, (b) FC3 in SNOWPACK RE-model, (c) FC1 in SNOWPACK DDA-model (d) FC3 in SNOWACK DDA-model, (e) FC1 in 3D model, and (f) FC3 in 3D model. Six profiles are shown in units of the arrival time at snow base in experiment, t (see Table 2).**

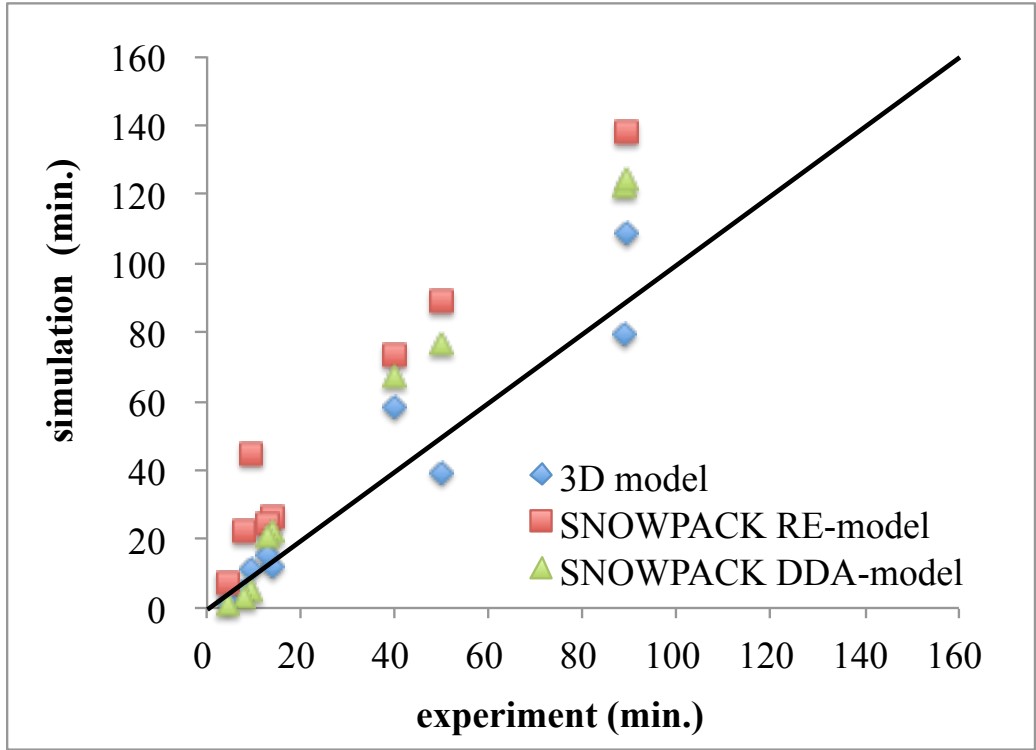

**Figure 7: Comparison of water arrival between laboratory experiment, a 3D-model and two SNOWPACK models.**

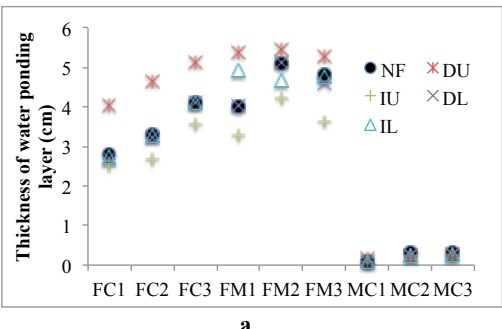

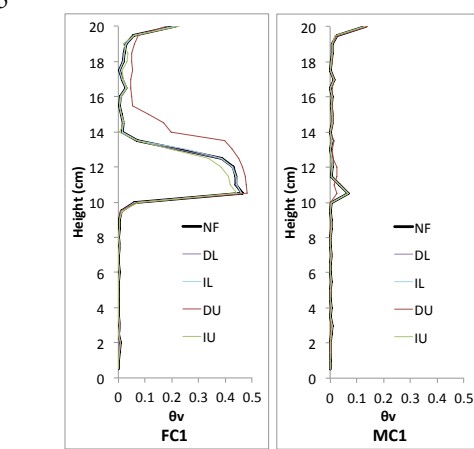

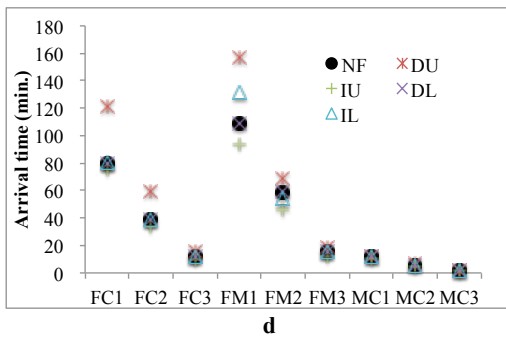

**Figure 8: Influence of grain size fluctuations on the thickness of the water ponding layer (a), water content profiles (b and c) and arrival time at the snow base (d). NF: no fluctuation, IU: increase 0.1 mm for the upper layer, DU: decrease 0.1 mm for the upper layer, IL: increase 0.1 mm for the lower layer and DL: decrease 0.1 mm for the lower layer.**

