# Peer review of "Liquid water infiltration into a layered snowpack: evaluation of a 3D water transport model with laboratory experiments"

_Hydrology and Earth System Sciences, 2017_

## Referee Comment (RC1) · Anonymous Referee #1 · 15 Jun 2017

General: This paper shows a very interesting model analysis of water transport in a snow sample. Both the experimental data and the model have been presented earlier and this paper now compares the model with the data set. This new comparison offers interesting insight into the process and offers a technically correct analysis. The presentation is concise, state of the art is presented in a balanced way and conclusions supported by the analysis. The paper is in scope and quality suitable for HESS.

One major suggestion concerns the SNOWPACK simulations. I understand that the reason for using the SNOWPACK version without preferential flow parameterization in order to show the differences caused by the preferential flow. However, I don't under-

stand why not additional simulations are shown, which use the SNOWPACK preferential flow parameterization. This would add value to the paper and increase its impact for two reasons: i) The 3D model is (computationally) limited to small domains (see also next comment below) and if the reader is provided with an analysis that allows to judge how much of the effect is covered by the SNOWPACK parameterization, then this has a lot of practical value for scientist that need to do larger-scale simulations. ii) The analysis would already give a first indication on how the slightly different treatment of preferential flow path initiation (entry suction) in the 3D model vs. SNOWPACK compares and would therefore add substance to the discussion of the entry suction problem, which is well executed in the paper otherwise. My suggestion is further supported by the fact that the current paper has not already too many new elements or this too long and this additional analysis should be easy to execute. One additional (major) comment concerns the missing discussion of the domain size effect.

While domain sizes have been chosen congruently between measurements and simulations, the generalization of the results may still be suffering from the small lateral extent in both. The dye experiments we know from snow (e.g. the ones from Schneebeli mentioned in the paper) show very significant lateral spreading of flow paths much beyond the scale of the experiments presented here. This aspect should be properly discussed.

Detailed (minor) comments:

p.3 l.27ff: Maybe mention additional snow characteristics (grain type) and how you produced the snow samples?

p.4 l.14: "of" breakthrough on breakthrough

p.4 l.19: Maybe grid points or elements instead of "meshes"

p.4 l.21: "expended" ?

p.4 l.29: "anticipated" measured

p.4 l.32: Here you could add some discussion on the effect of domain size (see above)

p.5 l.4: "85" add minutes

p.5 l.7ff: By comparing to Walter I think you compare two different things: Velocity of water in an existing preferential flow path (Walter) vs. propagation speed of path initiation. Please discuss appropriately

p.5 l.14: Please mention the reason of why it could not be determined

p.7 l.1ff: I would suggest to add a general comment that deviations of both models from measurements have the same order of magnitude

p.7 l.21: Why "in practice"?

p.8 l.8: "too vertically strict" is a funny expression and I suggest to try to explain it (maybe domain size as mentioned above?)

p.8 l.13ff: Here you should discuss that the model does not predict any structural changes in the snow grains. This is discussed further below but this is not sufficient. In reality, grains will grow quickly in contact with water at the walls of preferential flow paths and this will also promote lateral spreading of water, I think

p.9 l.4ff: Metamorphism could help to explain lateral spreading

Figure 1 legend: Explain the term "front grid"

---

## Referee Comment (RC2) · Anonymous Referee #2 · 12 Jul 2017

This paper presents model simulations performed with a recently developed 3D model for water movement through snow (published by the authors in 2014), to simulate recently published detailed experiments by Avanzi et al., 2016. These simulations show good agreement with the experimental results, for water content, timing, preferential flow, and ponding of water at capillary barriers. The results demonstrate that the recently developed model represents the complex physics of unsaturated flow in snow quite well, as long as the snow material properties (grain size, density, etc) are known.

There are a few additional steps that the authors could likely easily take, which would greatly improve this manuscript. I would highly recommend these improvements before

publication:

1) The authors compare their simulations to simulations using the operational SNOW-PACK model, and argue that their representation of preferential flow allows more accurate estimates of water flow timing and liquid water content. However, they use the SNOWPACK model without the recent improvements in this area (e.g. Wever et al, 2016; Wurzer et al, 2017), although these papers are cited. To complete this study, they should compare their small scale 3-D model with the most recent SNOWPACK model, which includes these improvements.

2) A sensitivity analysis would be very helpful. The authors use the measured grain sizes (from seiving) used in the experiment, and report these values to 0.001 mm. In nature, grain sizes are typically measured to 0.1mm, and models of grain growth are unlikely to be accurate to better than 0.5mm. It would be very helpful to see how sensitive their model results are to variations in grain size on the order of 0.1mm.

3) A recent similar paper using a different model, by a different group, also used the Avanzi et al, 2016 experiments and attempted to reproduce their results with a 2-D snow heat and mass flow model: Nicolas R. Leroux , John W. Pomeroy , Modelling capillary hysteresis effects on preferential ßow through melting and cold layered snow-packs, Advances in Water Resources (2017), doi:10.1016/j.advwatres.2017.06.024.

While the Leroux and Pomeroy paper was not published at the time this paper was submitted, now that it is accepted and online, it would be useful for the authors to cite and discuss the differences between their model and this one.

4) A more thorough discussion of model resolution would be appropriate. At the resolution of this model, it is unlikely modeling could be performed at the basin scale. How do the authors envision this new understanding of liquid water movement in snow, to impact large scale snow models? How could this understanding be implemented (emperically?) in operational modeling contexts?
Overall, this paper is well-written, presents a new result, and is suitable for publication in HESS.

Detailed line-by-line edits/suggestions are in the attached PDF.

Please also note the supplement to this comment:
https://www.hydrol-earth-syst-sci-discuss.net/hess-2017-200/hess-2017-200-RC2-supplement.pdf

**Supplement:**

[revised manuscript text omitted]

**Page: 2**

|   | T Autho   | r: reviewer2       | Subject: Inserted Text                                             | Date: 7/11/17, 1:51:51 PM                                       |
|---|-----------|--------------------|--------------------------------------------------------------------|-----------------------------------------------------------------|
| _ | Eirikss   | on, D., Whitson, N | I., Luce, C. H., Marshall, H. I                                    | P., Bradford,                                                   |
|   | J., Ber   | ner, S. G., Black, | T., Hetrick, H., and McNama                                        | ara, J. P.:                                                     |
|   | at hills  | lope and catchmer  | ologic relevance of lateral light
of scales. Hydrol. Process. 2 | 0w in show
27 640–654                                        |
|   | doi:10    | .1002/hyp.9666, 20 | )13.                                                               |                                                                 |
|   | A         |                    | Cubinet: Creen Out                                                 | Dete: 7/44/47 4-54-54 DM                                        |
| ~ | - T Autho | r: reviewerz       | Subject: Cross-Out                                                 | Date: 7/11/17, 1:51:51 PM                                       |
|   |           |                    |                                                                    |                                                                 |
|   | Autho     | r: reviewer2       | Subject: Highlight                                                 | Date: 7/11/17, 1:51:51 PM                                       |
| _ | Weve      | r et al, 2015 is a | different group of authors                                         | s, and this sentence applies to that paper, not the 2014 paper. |
|   |           | r: roviowor?       | Subject: Inserted Text                                             | Date: 7/11/17 1:51:51 PM                                        |
|   | Weve      | r et al (2015)     |                                                                    |                                                                 |
|   |           | · • un (2010)      |                                                                    |                                                                 |
|   | T Autho   | r: reviewer2       | Subject: Inserted Text                                             | Date: 7/11/17, 1:51:51 PM                                       |
|   | snowp     | backs typically co | ontain                                                             |                                                                 |
|   | _ Autho   | r: reviewer2       | Subject: Cross-Out                                                 | Date: 7/11/17. 1:51:51 PM                                       |
|   |           |                    | ,                                                                  |                                                                 |
| / | • ··      |                    | 0.11 × 1 × 1 <del>*</del> ×                                        |                                                                 |
| _ | T Autho   | r: reviewer2       | Subject: Inserted Text                                             | Date: 7/11/17, 1:51:51 PM                                       |
|   | S         |                    |                                                                    |                                                                 |
|   | T Autho   | r: reviewer2       | Subject: Inserted Text                                             | Date: 7/11/17, 1:51:51 PM                                       |
| / | a         |                    |                                                                    |                                                                 |
|   |           | r: roviowor?       | Subject: Inserted Text                                             | Date: 7/11/17 1:51:51 DM                                        |
| _ |           | I. IEVIEWEIZ       | Subject. Inserted Text                                             |                                                                 |
|   |    |                    |                                                                    |                                                                 |

[revised manuscript text omitted]

**Page: 4**

after

| _   | -                                                                          |                                                                               |                                                                                                                                          |
|-----|----------------------------------------------------------------------------|-------------------------------------------------------------------------------|------------------------------------------------------------------------------------------------------------------------------------------|
| _1  | Author: reviewer2                                                          | Subject: Highlight                                                            | Date: 7/11/17, 1:51:51 PM                                                                                                                |
|     | Doesn't make sense to
accurately, except with
Suggest reporting 2.9m | have 3 significant digits
CT scans, and the varial
m here, for example. | in the grain size - it is not possible to measure snow grain size this
bility between grains will always be much larger than 0.001mm. |
| F   | Author: reviewer2                                                          | Subject: Inserted Text                                                        | Date: 7/11/17, 1:51:51 PM                                                                                                                |
| -   | began                                                                      |                                                                               |                                                                                                                                          |
| 1   | Author: reviewer2                                                          | Subject: Inserted Text                                                        | Date: 7/11/17, 1:51:51 PM                                                                                                                |
|     | S                                                                          |                                                                               |                                                                                                                                          |
| J   | Author: reviewer2                                                          | Subject: Inserted Text                                                        | Date: 7/11/17, 1:51:51 PM                                                                                                                |
|     | s                                                                          |                                                                               |                                                                                                                                          |
| 1   | Author: reviewer2                                                          | Subject: Inserted Text                                                        | Date: 7/11/17, 1:51:51 PM                                                                                                                |
| - 🖻 | s                                                                          |                                                                               |                                                                                                                                          |
| F   | Author: reviewer2                                                          | Subject: Cross-Out                                                            | Date: 7/11/17, 1:51:51 PM                                                                                                                |
|     |                                                                            |                                                                               |                                                                                                                                          |
|     |                                                                            |                                                                               |                                                                                                                                          |

Author: reviewer2 Subject: Inserted Text Date: 7/11/17, 1:51:51 PM

Hirashima et al. (2014a). A smaller path area would increase conductivity because liquid water would be more concentrated and push water towards the boundary faster. After arrival at the boundary, we found that liquid water ponded above the boundary owing to a capillary barrier. In images from just before the formation of preferential flow in the lower layer (Fig. 1c and g), the elapsed time was  $85_{1}$  in the laboratory experiment and 79 minutes in the simulation (relative difference of 7%

- 5 of the measurement value, i.e., in good agreement). The times of liquid water arrival at the base following the formation of preferential flow through the lower layer were 4 and 1 minutes in the laboratory experiment and simulation, respectively. On this basis, we calculated the propagation rate of the preferential flow path to be 0.4 and 1.6 mm/s for the laboratory experiment and simulation, respectively. This is one order of magnitude smaller than mean speed of preferential flow (11.2 mm/s) measured by Walter et al. (2013) using Duorescent
- 10 particle tracking velocimetry; however, this was tested with a much larger water supply rate (3600 much). In the other experiments, the temporal dynamics of preferential flow formation and water ponding at the interface were generally well reproduced (Table 2; Fig. 2). The Root Mean Square Error, the slope of a regression line with intercept equal to 0, and the correlation coefficient (r2) between the simulated and measured timings were 7.8 min., 0.97 and 0.93, respectively. As timings were measured using frontal movies, we were sometimes unable to evaluate the timing of
- 15 preferential flow formation within a sample (e.g., experiment MC1); therefore, estimated timings from laboratory experiment may contain a delay. Overall, simulated and measured timings coincided well, which confirms that if snow parameters (e.g., snow density and grain size) are known, the arrival time of liquid water can be predicted using this model.

**3.2 Thickness of water ponding layer**

- 20 Avanzi et al. (2016) measured the thickness of the upper layer affected by ponding at the end of each experiment. In their results for FC and FM experiments, the liquid water content on the layer boundary was about 33–36% (2-cm vertical resolution of data). The volume of ponded water was smaller for MC experiments. Laboratory experiments confirmed that the thickness of the water ponding layer is not strongly connected to the water input rate. Here, the influence of water input rate on the thickness of water ponding layer was also small; however, the influence of grain size was significant (Table 3).
- 25 The thickness of ponded water at the interface was well reproduced for the FC experiments, but was overestimated for FM experiments and underestimated for MC experiments. For MC experiments, up to 1 cm of ponding was shown in laboratory experiments, while simulated results showed a thickness of less than 0.5 cm.

**3.3 Horizontal cross section**

30 During laboratory experiments, Avanzi et al. (2016) measured wet snow fractions at the boundary between consecutive rings using photos of the top surface of the ring below the boundary. Samples were likely slightly compressed during experiments, (Marshall et al. 1999), even though this was not noticeable. Because the model does not include settling, we chose to

5

**Page: 5**

| T | Author: reviewer2 | Subject: Inserted Text | Date: 7/11/17, 1:51:51 PM |
|---|-------------------|------------------------|---------------------------|
|   | min               |                        |                           |
| Т | Author: reviewer2 | Subject: Inserted Text | Date: 7/11/17, 1:51:51 PM |
| ~ | (RMSE)            |                        |                           |

Author: reviewer2 Subject: Inserted Text Date: 7/11/17, 1:51:51 PM due to increased densification caused by wetting 
[revised manuscript text omitted]

---

## Author Comment (AC2) · 18 Aug 2017

Response for Reviewer 2

This paper presents model simulations performed with a recently developed 3D model for water movement through snow (published by the authors in 2014), to simulate recently published detailed experiments by Avanzi et al., 2016. These simulations show good agreement with the experimental results, for water content, timing, preferential flow, and ponding of water at capillary barriers. The results demonstrate that the recently developed model represents the complex physics of unsaturated flow in snow quite well, as long as the snow material properties (grain size, density, etc) are known.

There are a few additional steps that the authors could likely easily take, which would greatly improve this manuscript. I would highly recommend these improvements before publication:

1) The authors compare their simulations to simulations using the operational SNOW-PACK model, and argue that their representation of preferential flow allows more accurate estimates of water flow timing and liquid water content. However, they use the SNOWPACK model without the recent improvements in this area (e.g. Wever et al, 2016; Wurzer et al, 2017), although these papers are cited. To complete this study, they should compare their small scale 3-D model with the most recent SNOWPACK model, which includes these improvements.

Response: Thank you for constructive comments. The first reviewer also suggested to add SNOWPACK simulations with the dual domain approach and we agree with both of you that including this scheme would enhance the impact of the work. We have already performed such simulations, which will be included in our revised manuscript. The variables of interest of this new comparison will be water content profiles and liquid water arrival at the snow base. The discussion section will be modified as suggested.

2) A sensitivity analysis would be very helpful. The authors use the measured grain sizes (from seiving) used in the experiment, and report these values to 0.001 mm. In nature, grain sizes are typically measured to 0.1mm, and models of grain growth are unlikely to be accurate to better than 0.5mm. It would be very helpful to see how sensitive their model results are to variations in grain size on the order of 0.1mm.

Response: We agree with you. We will perform additional simulations with changes in grain size in the order of 0.1 mm for both layers. A discussion about changes in thickness of water ponding layer, water content distribution, and arrival time at sample base will be then included in the manuscript.

3) A recent similar paper using a different model, by a different group, also used the Avanzi et al, 2016 experiments and attempted to reproduce their results with a 2-D
snow heat and mass flow model: Nicolas R. Leroux , John W. Pomeroy , Modelling capillary hysteresis effects on preferential flow through melting and cold layered snowpacks, Advances in Water Resources (2017), doi:10.1016/j.advwatres.2017. 06.024. While the Leroux and Pomeroy paper was not published at the time this paper was submitted, now that it is accepted and online, it would be useful for the authors to cite and discuss the differences between their model and this one.

Response: In the revised manuscript, we will consider the Leroux and Pomeroy paper and add a discussion about the difference between those and our results as follows:

"The main difference between the two models is the number of dimensions considered, i.e., 2D in Leroux and Pomeroy (2017) and 3D (this paper). While Leroux and Pomeroy's model also include temperature and melt-freeze processes, this is not expected to play a role here as the validation experiments were performed in isothermal conditions. In natural snow, water flow shows lateral spreading, especially at capillary barriers, which creates complex three-dimensional stratigraphic features at grain/layer scale. Furthermore, when 3D preferential flow paths form in dry snow, wet snow area is proportional to the square of preferential flow size and inversely proportional to the square of the distance between paths (see Fig. S1). In case of a two-dimensional simulation, wet snow area will be, on the contrary, proportional to preferential flow size and inversely proportional to the distance between paths (see Fig. S1). Considering a 3D geometry can, therefore, help to define the necessary parameterizations of preferential flow effects needed to inform models with a reduced number of dimensions." This discussion is supported by the figure, which will be added as Fig. S1 in the Supplement. (see attached supplement file)

4) A more thorough discussion of model resolution would be appropriate. At the resolution of this model, it is unlikely modeling could be performed at the basin scale. How do the authors envision this new understanding of liquid water movement in snow, to impact large scale snow models? How could this understanding be implemented (emperically?) in operational modeling contexts?

Response: Thank you for this important advice. We agree with you that the scale of this model is still not suitable for direct applications at the basin scale. We will include a specific discussion about this in sub-section 4.3, or in an additional sub-section about future work.

"Our results show that this model is capable of reproducing detailed water infiltration at sample scale, i.e., considering micro-scale heterogeneity. On the other hand, the intrinsic scale of this process and computational efforts make it still not suitable for basin-scale simulations. This limitation could be overcome by synergies with existing physics-based hydrologic models for snow-dominated catchments, e.g., Alpine 3D (Lehning et al., 2006). Currently, SNOWPACK is used as a part of Alpine3D for simulation of accumulation/ablation patterns of snowpack. In this paper, comparison between laboratory experiments, a 3D model and SNOWPACK were performed and contributed to highlight model limitations and possible avenues of future developments, e.g., an underestimation of flow path cross-sections. While a 3D model cannot reproduce the entire range of natural variability of liquid water flow in snow, it can help to replicate and understand this process in conditions that are difficult for experiments, e.g., larger sample sizes and/or a more complex stratigraphy. This may contribute to define new parameterizations for dual domain approaches that could be then fully included in catchment-scale models. Also, we will try to apply this model at the basin scale by increasing the element size. While this will hamper the representation of single preferential fingers, we expect the model to be able to correctly reproduce other relevant features of water flow at slope scale such as lateral flow. This could help to understand liquid water flow around concave/convex portions of the landscape."

Detailed line-by-line edits/suggestions are in the attached PDF. Please also note the supplement to this comment: https://www.hydrol-earth-syst-sci-discuss.net/hess-2017-200/hess-2017-200-RC2- supplement.pdf

Response: Thank you for detailed suggestions. We have welcomed your suggestions in the manuscript and are going to address your comments in the revised paper.

Please also note the supplement to this comment:
https://www.hydrol-earth-syst-sci-discuss.net/hess-2017-200/hess-2017-200-AC2-supplement.pdf

———————————————————

[Figure]

**Supplement:**

[Figure]

Fig. S1 Effect of the number of dimensions of a model on simulated wet snow area. Figures S1 (a), (c), (e) and (g) refer to a 3D snow section with size 10 x 10 cm$^2$; Figures S1 (b), (d), (f) and (h) refer to a 2D snow section with size 10cm. The initial conditions (Figures S1c and S1d) are preferential flow paths of size equal to 1 cm and distance between paths equal to 5 cm. If the width of 3D preferential flow paths doubles, the ratio of wet snow area will become four times higher because both sides of the paths double (e). On the other hand, in a 2D simulation, wet snow area will double (f). If the distance between preferential flow paths reduces to 2.5 cm, the number of paths will become four times larger in a 3D simulation, while it will double in a 2D simulation.

---

## Author Response (AR1)

Response for reviewer 1

*General: This paper shows a very interesting model analysis of water transport in a snow sample. Both the experimental data and the model have been presented earlier and this paper now compares the model with the data set. This new comparison offers interesting insight into the process and offers a technically correct analysis. The presentation is concise, state of the art is presented in a balanced way and conclusions supported by the analysis. The paper is in scope and quality suitable for HESS.*

*One major suggestion concerns the SNOWPACK simulations. I understand that the reason for using the SNOWPACK version without preferential flow parameterization in order to show the differences caused by the preferential flow. However, I don't understand why not additional simulations are shown, which use the SNOWPACK preferential flow parameterization. This would add value to the paper and increase its impact for two reasons: i) The 3D model is (computationally) limited to small domains (see also next comment below) and if the reader is provided with an analysis that allows to judge how much of the effect is covered by the SNOWPACK parameterization, then this has a lot of practical value for scientist that need to do larger-scale simulations.*

*ii) The analysis would already give a first indication on how the slightly different treatment of preferential flow path initiation (entry suction) in the 3D model vs. SNOWPACK compares and would therefore add substance to the discussion of the entry suction problem, which is well executed in the paper otherwise. My suggestion is further supported by the fact that the current paper has not already too many new elements or this too long and this additional analysis should be easy to execute.*

Response: Thank you for these constructive comments. The second reviewer also suggested to add SNOWPACK simulations with the dual domain approach and we agree with both of you that including this scheme would enhance the impact of the work. We therefore included SNOWPACK with dual-domain approach in our approach (SNOWPACK DDA-model). Also, SNOWPACK without preferential flow is now named SNOWPACK RE-model to distinguish it from DDA-model. The resolution of SNOWPACK simulations was changed to 5mm to match with the resolution of 3D model. Thickness of water ponding layer, water content profiles and liquid water arrival

at the snow base, were compared. These discussions are reported in section 4.1. Figure 6 in revised manuscript now includes the water profile of DDA-model. Also, a comparison figure about arrival time for models was added as Figure 7.

*One additional (major) comment concerns the missing discussion of the domain size effect. While domain sizes have been chosen congruently between measurements and simulations, the generalization of the results may still be suffering from the small lateral extent in both. The dye experiments we know from snow (e.g. the ones from Schneebeli mentioned in the paper) show very significant lateral spreading of flow paths much beyond the scale of the experiments presented here. This aspect should be properly discussed.*

Response: We agree with you. In the revised manuscript, we will add a discussion about the domain size effect. This discussion is here reported as a reply for minor comment at P4 l32 (see below)

*Detailed (minor) comments:*

*p.3 l.27ff: Maybe mention additional snow characteristics (grain type) and how you produced the snow samples?*

Response: The details about production of snow samples are already reported in Avanzi et al (2016). However, we will add some information about the preparation method and grain type.

"In these experiments, snow samples were prepared in a cold room at −20∘C using refrozen melt forms. Snow was packed in a cylindrical container composed of several acrylic rings; the height and diameter of the rings were 20 mm and 50 mm, respectively. Each sample was composed of two layers: the upper layer was 10 cm thick, the lower layer was either 8 or 10 cm thick (see Avanzi et al., 2016). Then, samples were moved to a second cold room at 0∘C and stored for at least 12 h to reach initial conditions of dry snow at 0∘C. All samples were characterized by a finer-over-coarser layering (i.e. the upper layer was created using a smaller grain size than the lower one), which aimed

to reproduce capillary barriers. The three classes of snow grain size included fine (0.25–0.5 mm), medium (1.0–1.4 mm), and coarse (2.0–2.8 mm)."

*p.4 l.14: "of" breakthrough on breakthrough*

Response: We clarified the sentence as reported below:

"The evaluation of simulations focused on the thickness of the ponding layer at the textural interface, on the liquid water distribution, on the wet snow fraction at different heights, and on different timings that are relevant for liquid water movement in snow. These include water arrival at the interface between layers, breakthrough of preferential flow in the lower layer, and arrival of liquid water at sample base."

*p.4 l.19: Maybe grid points or elements instead of "meshes"*

Response: We clarified the sentence as reported below. Also, in revised manuscript, we will check words and replace, "meshes" with "elements" in the entire manuscript.

"The simulated timings of water arrival at the interface, entering the lower layer, and arrival at the snow base refer to the lowest elements in the upper layer, the top three elements in lower layer, and the lowest elements of the sample, respectively."

*p.4 l.21: "expended" ?*

Response: We clarified the sentence as reported below:

"The water content in the top three elements of the lower layer was used to determine the timing of breakthrough because preferential flow began immediately after the water content of one of these elements became larger than zero."

*p.4 l.29: "anticipated" measured*

Response: We clarified the sentence as reported below:

"Some images of the development of capillary barriers and preferential flow for FC1 are shown in Fig. 1. These figures show the front surfaces 20 s after the beginning of the experiment (a and e), at the arrival time of water at the interface between layers (b and f), at the time of breakthrough of preferential flow into the lower layer (c and g) and at the arrival time at the snow base (d and h). The simulation results showed faster than measured arrival of water at the boundary (Table 2), which implied an overestimation of vertical velocity in the model's preferential flow for this experiment (Fig. 1)."

*p.4 l.32: Here you could add some discussion on the effect of domain size (see above)*

Response: We agree with you and have included some discussion on this point as reported below:

"The size of laboratory experiments was restricted by the time needed to prepare and perform each of them. Also, the diameter of samples was consistent with the thickness of similar experiments in soils (see e.g. Hill and Parlange, 1972), whereas Avanzi et al. (2017) showed that preferential flow may be intrinsically coupled with wet-snow metamorphism at grain scale. This suggests that small-scale experiments are appropriate to understanding the physics of this process in snow. Nonetheless, the relatively small scale of these experiments may introduce some domain size effect. In natural snow, water flow shows lateral spreading, especially at capillary barriers, whereas experiments with small size may partially perturb the natural flow on snow and therefore change vertical flow owing to artificial edges. This may increase the ratio of preferential flow path area, decrease the arrival time at the base, and decrease natural ponding amount at the capillary barrier. In terms of comparison between simulations and experiments, this effect was offset by using the same domain conditions."

*p.5 l.4: "85" add minutes*

Response: This information was added to the manuscript.

*p.5 l.7ff: By comparing to Walter I think you compare two different things: Velocity of*

*water in an existing preferential flow path (Walter) vs. propagation speed of path initiation. Please discuss appropriately*

Response: As you suggested, the velocity in an existing preferential flow path is likely different from propagation speed of path initiation. This difference makes this comparison unsuitable. Therefore, we removed the sentence "This is one order of magnitude …. supply rate (3600 mm/h). " from the manuscript. Thank you.

*p.5 l.14: Please mention the reason of why it could not be determined*

Response: We were not be able to measure the exact timing of preferential flow initiation using a frontal movie when preferential flow formed at the posterior side of samples with respect to the position of the camera. On the other hand, horizontal spreading of water at sample base always allowed to detect the arrival time with a reasonable precision. We clarified this in the manuscript.

"As timings were measured using frontal movies, we were sometimes unable to evaluate the timing of preferential flow formation within a sample. For example, in the case of MC1, preferential flow formed on the side of the sample that was not visible from the frontal position of the camera. Thus, the frontal movie did not show the preferential flow path in lower layer, whereas horizontal spreading of water at the sample base allowed us to detect the arrival time with a reasonable precision. It follows that for this sample we cannot determine the timing of preferential flow initiation. Therefore, estimated timings from laboratory experiments may contain a delay."

*p.7 l.1ff: I would suggest to add a general comment that deviations of both models from measurements have the same order of magnitude*

Response: We quantified the magnitude of deviation for each simulation. While both SNOWPACK schemes yield the same RMSE, the 3D model returns a slightly smaller value. So we described about deviations as follows.

"The RMSE for liquid water content profiles at (measured) arrival time are 0.107, 0.107

and 0.094 for SNOWPACK RE-model, DDA-model and 3D model, respectively. While both SNOWPACK schemes yield the same RMSE, the 3D model returns a slightly smaller value."

*p.7 l.21: Why "in practice"?*

Response: "In practice" is not suitable to use here. So it was removed.

*p.8 l.8: "too vertically strict" is a funny expression and I suggest to try to explain it (maybe domain size as mentioned above?)*

Response: As we replied to your major comment, we expect domain size effects to possibly perturb real dynamics of liquid water flow by increasing wet snow area. This effect is, nonetheless, in contrast with the observed underestimation of wet snow area by the model. We elaborated on this sentence and connected it to the next paragraph.

"Nevertheless, the simulated mean wet snow area was small even for fine snow (e.g. 4.8% in FC1 and 22% in FC3, excluding the ponding area). As the simulated wet snow area is smaller than the measured one, this model may still underestimate the effective cross-sectional area of infiltration. This will be the subject of future research."

*p.8 l.13ff: Here you should discuss that the model does not predict any structural changes in the snow grains. This is discussed further below but this is not sufficient. In reality, grains will grow quickly in contact with water at the walls of preferential flow paths and this will also promote lateral spreading of water, I think*

Response: Although this model includes grain growth following Brun et al. (1989) and Tusima (1977), modeling some specific conditions such as wet snow metamorphism at the boundaries between preferential flow paths and drier snow is still an open issue. This is important to improve the model. Thank you for this advice. We will add a discussion about it.

"Neglecting quick metamorphism in preferential flow paths (Avanzi et al. 2017) may

represent another cause of underestimation of preferential flow path size as grain growth promotes lateral spreading of water and expansion of paths. Although this model includes grain growth following Brun et al. (1989) and Tusima (1977), modelling some specific conditions such as wet snow metamorphism at the boundaries between preferential flow paths and drier snow is still an open issue. Also, existing observations of wet-snow metamorphism have been mainly performed in static conditions, which means that the coupling between grain growth and flowing water is still poorly understood. This represents a further unknown for models of liquid water in snow."

*p.9 l.4ff: Metamorphism could help to explain lateral spreading*

Response: We agree with your opinion. We will add the following discussion at the end of Section 4.3.

"Extending the existing parameterizations of wet snow metamorphism for small timescales will improve simulation accuracy with regards to the development and disappearance of water ponding by capillary barriers. Also, a correct simulation of grain growth will lead to correctly estimating the lateral spreading of water, which will improve the accuracy of the prediction of preferential flow path size."

*Figure 1 legend: Explain the term "front grid"*

Response: We thought using phrase "front elements" is better than "front grid". So we replaced it in the manuscript. Also, we will improve the figure to explain the term "front elements" (see below). This will also be mentioned in the caption.

"Figure 1: Development of capillary barrier and preferential flow path for FC1 during experiments (a–d) and simulations (e–h). A blue dye tracer was used in the experiment. In the simulation images, blue denotes the liquid water content at the front elements (see: right figure), while grey denotes that the front elements are dry, but some liquid water is present within the sample at that position. The grey-scale represents the maximum liquid water content for each location. Captured times were at: (a) 20 s, (b) 35 min, (c) 1 hour 25 min, (d) 1 hour 29 min, (e) 20 s, (f) 17 min, (g) 1 hour 19 min,

and (h) 1 hour 20 min."

Response for Reviewer 2

*This paper presents model simulations performed with a recently developed 3D model for water movement through snow (published by the authors in 2014), to simulate recently published detailed experiments by Avanzi et al., 2016. These simulations show good agreement with the experimental results, for water content, timing, preferential flow, and ponding of water at capillary barriers. The results demonstrate that the recently developed model represents the complex physics of unsaturated flow in snow quite well, as long as the snow material properties (grain size, density, etc) are known. There are a few additional steps that the authors could likely easily take, which would greatly improve this manuscript. I would highly recommend these improvements before publication:*

*1) The authors compare their simulations to simulations using the operational SNOWPACK model, and argue that their representation of preferential flow allows more accurate estimates of water flow timing and liquid water content. However, they use the SNOWPACK model without the recent improvements in this area (e.g. Wever et al, 2016; Wurzer et al, 2017), although these papers are cited. To complete this study, they should compare their small scale 3-D model with the most recent SNOWPACK model, which includes these improvements.*

Response: Thank you for these constructive comments. The first reviewer also suggested to add SNOWPACK simulations with the dual domain approach and we agree with both of you that including this scheme would enhance the impact of the work. We therefore included SNOWPACK with dual-domain approach in our discussion (SNOWPACK DDA-model). Also, SNOWPACK without preferential flow is now named SNOWPACK RE-model to distinguish it from DDA-model. The resolution of SNOWPACK simulations was changed to 5mm to match with the resolution of 3D model. Thickness of water ponding layer, water content profiles and liquid water arrival at the snow base, were compared. These discussions are reported in section 4.1. Figure 6 in revised manuscript now includes the water profile of DDA-model. Also, a comparison figure about arrival time for models was added as Figure 7.

*2) A sensitivity analysis would be very helpful. The authors use the measured grain sizes (from seiving) used in the experiment, and report these values to 0.001 mm. In nature, grain sizes are typically measured to 0.1mm, and models of grain growth are unlikely to be accurate to better than 0.5mm. It would be very helpful to see how sensitive their model results are to variations in grain size on the order of 0.1mm.*

Response: We agree with you. We performed additional simulations with changes in grain size in the order of 0.1 mm for both layers. A new sub-section (4.3) was added to discuss changes in thickness of water ponding layer, water content distribution, and arrival time at sample base due to fluctuations of grain sizes. Figure 8 and Figure S5 were also added to show the result of sensitivity experiment. Sensitivity experiments showed large influence of grain size fluctuation of upper fine snow and suggested the importance of careful measurement of grain size for fine snow.

*3) A recent similar paper using a different model, by a different group, also used the Avanzi et al, 2016 experiments and attempted to reproduce their results with a 2-D snow heat and mass flow model: Nicolas R. Leroux , John W. Pomeroy , Modelling capillary hysteresis effects on preferential flow through melting and cold layered snowpacks, Advances in Water Resources (2017), doi:10.1016/j.advwatres.2017.06.024.*
*While the Leroux and Pomeroy paper was not published at the time this paper was submitted, now that it is accepted and online, it would be useful for the authors to cite and discuss the differences between their model and this one.*

Response: In the revised manuscript, we will consider the Leroux and Pomeroy paper and add a discussion about the difference between those and our results as follows:

"Similarly, Leroux and Pomeroy (2017) developed a 2D water transport model basing on the scheme of Hirashima et al. (2014a), but considering melt-freeze processes. Reproducing heterogeneous processes in a 1D or 2D model requires several assumptions. In natural snow, water flow shows lateral spreading, especially at capillary barriers, which creates complex 3D stratigraphic features at a grain/layer scale. Furthermore, when 3D preferential flow paths form in dry snow, wet snow area is

proportional to the square of preferential flow size and inversely proportional to the square of the distance between paths (see Fig. S1). For a 2D simulation, wet snow area is, e.g., proportional to preferential flow size and inversely proportional to the distance between paths (see Fig. S1). Considering a 3D geometry can, therefore, help to define the necessary parameterizations of preferential flow effects needed to inform models with a reduced number of dimensions. Note that, while Leroux and Pomeroy's model also includes temperature and melt-freeze processes, this is not expected to play a role here as the validation experiments were performed under isothermal conditions. "

This discussion is supported by the figure, which was added as Fig. S1 in the Supplement.

*4) A more thorough discussion of model resolution would be appropriate. At the resolution of this model, it is unlikely modeling could be performed at the basin scale. How do the authors envision this new understanding of liquid water movement in snow, to impact large scale snow models? How could this understanding be implemented (emperically?) in operational modeling contexts?*

Response: Thank you for this important advice. We agree with you that the scale of this model is still not suitable for direct applications at the basin scale. We will include a specific discussion about this in sub-section 4.3, or in an additional sub-section about future work.

"Our results show that this model is capable of reproducing detailed water infiltration at sample scale (i.e., considering micro-scale heterogeneity). On the other hand, the intrinsic scale of this process and computational efforts mean that it is still not suitable for basin-scale simulations. This limitation could be overcome by synergies with existing physics-based hydrologic models for snow-dominated catchments; for example, Alpine 3D (Lehning et al., 2006). Currently, SNOWPACK is used as a part of Alpine3D for simulation of accumulation/ablation patterns of snowpack. In this study, comparisons between laboratory experiments, a 3D model, and SNOWPACK were performed and contributed to highlighting model limitations and possible avenues of future developments (e.g. an underestimation of flow path cross-sections). While a 3D

model cannot reproduce the entire range of natural variability of liquid water flow in snow, it can help to replicate and understand this process in conditions that are difficult for experiments (e.g., larger sample sizes and/or a more complex stratigraphy). This may contribute to defining new parameterizations for dual domain approaches that could be then fully included in catchment-scale models. Also, we will try to apply this model at the basin scale by increasing the element size. While this will hamper the representation of single preferential fingers, we expect the model to be able to correctly reproduce other relevant features of water flow at slope scale such as lateral flow. This could help to understand liquid water flow around concave/convex portions of the landscape. "

*Detailed line-by-line edits/suggestions are in the attached PDF.*
*Please also note the supplement to this comment: https://www.hydrol-earth-syst-sci-discuss.net/hess-2017-200/hess-2017-200-RC2-* supplement.pdf

Response: Thank you for detailed suggestions. We have welcomed your suggestions in the manuscript and are going to address your comments in the revised paper.

[revised manuscript text omitted]